# MAKING AND EVALUATING CALIBRATED FORECASTS

## ABSTRACT

Calibrated predictions can be reliably interpreted as probabilities. An important step towards achieving better calibration is to design an appropriate calibration measure to meaningfully assess the miscalibration level of a predictor. A recent line of work initiated by Haghtalab et al. (2024) studies the design of truthful calibration measures: a truthful measure is minimized when a predictor outputs the true probabilities, whereas a non-truthful measure incentivizes the predictor to lie so as to appear more calibrated. All previous calibration measures were non-truthful until Hartline et al. (2025) introduced the first perfectly truthful calibration measures for binary prediction tasks in the batch setting.

We introduce a perfectly truthful calibration measure for multi-class prediction tasks, generalizing the work of Hartline et al. (2025) beyond binary prediction. We study common methods of extending calibration measures from binary to multi-class prediction and identify ones that do or do not preserve truthfulness. In addition to truthfulness, we mathematically prove and empirically verify that our calibration measure exhibits superior robustness: it robustly preserves the ordering between dominant and dominated predictors, regardless of the choice of hyperparameters (bin sizes). This result addresses the non-robustness issue of binned ECE, which has been observed repeatedly in prior work.

## 1 INTRODUCTION

Calibration ensures that predictions can be reliably interpreted as probabilities (Dawid, 1982). For example, in weather forecasting, a predictor outputs a prediction $p \in [0, 1]$ of a binary outcome $y \in \{0, 1\}$, rainy or not rainy. Among the days that the predictor outputs $p = 40\%$ chance of rain, calibration requires the actual empirical frequency of rain $\Pr[y = 1 | p = 40\%]$ to be the same $40\%$. Formally, calibration is the requirement $\Pr[y = 1 | p] = p$ for every prediction value $p \in [0, 1]$.

To quantify the level of miscalibration, it is common practice to evaluate a predictor using a calibration measure. For example, one canonical calibration measure is the Expected Calibration Error (ECE). For a binary classification task, ECE is defined as the expected absolute prediction bias, $\mathbf{E}\left[\left|p - \Pr[y = 1 | p]\right|\right]$.

Recent work observes that known calibration errors prior to the work of Hartline et al. (2025) are not truthful (Qiao & Valiant, 2021; Haghtalab et al., 2024). Truthfulness of an error metric requires that the expected error is minimized when the Bayesian predictor outputs the true distribution of the outcome. The following example explains the non-truthfulness of ECE.

**Example 1.1.** *We estimate calibration error over $n$ samples. The ground truth distribution of the outcome $y$ for each sample $i$ is $\frac{i}{n}$, uniformly distributed in $[0, 1]$. The optimal predictor has* ECE $\sim O(1)$ *- when estimating the conditional probability $\Pr[y = 1 | p]$ for each $p = \frac{i}{n}$, each prediction corresponds to only one sample. However, a non-truthful predictor that always outputs $0.5$ achieves an* ECE *of $O(1/\sqrt{n})$.*

Binning, a common practice for estimating the conditional probability, also leads to non-truthful calibration measures. A binning strategy divides the prediction space into intervals and conditions on the bin instead of the prediction value. For example, the binned ECE divides the space of $[0, 1]$ into $m$ equal intervals. For the truthful predictor in Theorem 1.1, each interval contains $\frac{n}{m}$ predictions. The estimated bias is then $O\left(\sqrt{1/(n/m)}\right) = O\left(\sqrt{m/n}\right)$. For the constant predictor, only one interval contains $n$ constant predictions, still achieving an $O(1/\sqrt{n})$ error.

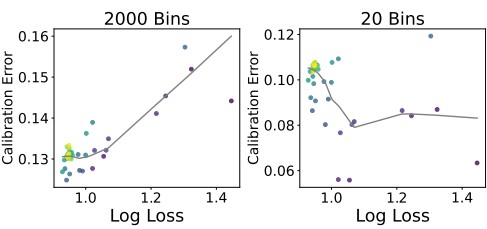 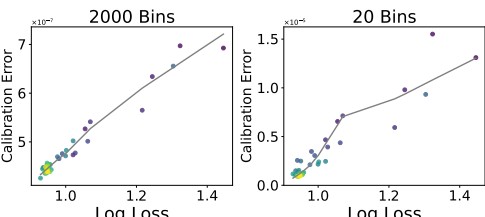

(a) $\ell_1$-QECE-confidence is **non-truthful**. The left estimates ECE with $m = 2000$ bins, and more accurate predictors have lower calibration error. The right estimates with $m = 20$ bins, and more accurate predictors have higher calibration error.[1]

(b) $\ell_2$-QECE-classwise is a **truthful** calibration error. The left plot estimates the calibration error with $m = 2000$ bins, and the right plot estimates with $m = 20$ bins. The ranking between predictors is more consistent for different numbers of bins.

Figure 1: We compare each calibration measure with different number of bins. Each dot in the plot is a predictor. The $x$-axis plots the log loss, while the $y$-axis plots a calibration error. Figure 1a replicates the result in Minderer et al. (2021).

As a result of the non-truthfulness, the rankings between the optimal (truthful) predictor and non-optimal predictors are not robust to this hyperparameter $m$, the number of bins. To see this, a calibrated predictor may have non-zero calibration error from the sampling error, and the quantification of sampling error (for the example above, the calibration error $\sqrt{m/n}$) depends on the number $m$ of bins. Nixon et al. (2019) and Minderer et al. (2021) observe this non-robustness - the ranking of predictors by binned ECE may flip when $m$ changes. This non-robustness leads to inconsistent conclusions about which predictor is more trustworthy (Figure 1a) and raises questions about how to select the number of bins.

Truthfulness has been the principle for eliciting and evaluating probabilistic predictions (Gneiting, 2011)[2]. In information elicitation, if a predictor is strategizing to minimize the expected error, a truthful error incentivizes the predictor to output the true distribution. When an error is used for training, selecting, or just comparing predictors, truthfulness aligns incentives with learning the true probabilities.

In statistical decision theory, a truthful error preserves the dominance ranking between predictors. With a truthful error, the expected error decomposes into an entropy term (depending only on the data-generating distribution) plus a Bregman divergence from the true distribution to the prediction. Hence, model rankings by expected score are exactly rankings by divergence from truth; lower proper loss implies closer to the truth and thus a better rank. Moreover, a truthful error is equivalent to the Bayes risk on a decision problem. If predictor $f$ *dominates* predictor $g$ across decision problems, i.e., achieves no larger Bayes risk for every decision problem. A truthful error also preserves decision-theoretic dominance and cannot invert a uniformly superior predictor.

In this work, we design truthful calibration errors for multi-class classification tasks and demonstrate (theoretically and empirically) that truthful errors robustly preserve the ranking between predictors.

**Theoretical Contributions.** We introduce a truthful calibration measure for multi-class prediction. We obtain this calibration measure by generalizing the truthful calibration measures for binary prediction from Hartline et al. (2025). A common practice for constructing multi-class calibration measures is to reduce the $k$-class problem to binary subproblems and aggregate the calibration errors in the subproblems. The most commonly used aggregation method is the confidence aggregation in Guo et al. (2017) (see Definition 2.8), which evaluates the calibration of the outcome with the highest predicted probability. However, we show that this approach does not in general result in a truthful calibration measure even when the binary calibration measure is truthful. In contrast, our Theorem 3.1 shows that, a different aggregation method, class-wise aggregation (Kull et al., 2019), preserves the truthfulness from binary prediction to multi-class prediction. Thus, we obtain our truthful multi-class calibration measure $\ell_2$-QECE$^{(\text{classwise})}$ by applying class-wise aggregation to the truthful measure $\ell_2$-QECE from Hartline et al. (2025).

---

[1]For $\ell_2$-QECE-confidence which is also non-truthful, the plots are almost the same as $\ell_1$-QECE-confidence.
[2]Truthfulness is also known as properness of a loss function (McCarthy, 1956; Savage, 1971).

In Section 4, we provide a theoretical justification for the robustness of our truthful calibration measure, which we demonstrate in Figure 1 and in our experiments in Section 5. Specifically, we prove Theorem 4.1 showing that for every pair of calibrated predictors $f_1, f_2$, if $f_1$ achieves smaller expected loss according to all proper losses (i.e., $f_1$ dominates $f_2$, see Definition 2.9), then $f_1$ also has smaller expected error using our truthful calibration measure $\ell_2\text{-QECE}^{(\text{classwise})}$. This *dominance-preserving property* holds independent of the hyperparameter (number of bins) used to compute $\ell_2\text{-QECE}^{(\text{classwise})}$. We also provide concrete examples in Section 4 showing that other non-truthful measures do not have the desired properties of dominance-preserving and robustness.

**Empirical Validation.** Our experiments demonstrate that the truthful calibration error preserves decision-task dominance between predictors and is robust to hyperparameter choices (i.e., number of bins). Section 5 evaluates a suite of neural network predictors under several strictly proper losses and observes a dominance between the predictors. The truthful calibration error preserves this ranking across models, and reported results remain stable across discretization settings (Figure 1b), unlike non-truthful metrics whose rankings can flip with the number or placement of bins.

### 1.1 RELATED WORK

**Truthful Calibration Measure.** Recent work introduced the notion of *truthfulness* for a calibration measure. Haghtalab et al. (2024) first formalize this concept and design an approximately truthful calibration measure in the online setting. Qiao & Zhao (2025) study calibration errors that are approximately truthful and quantify decision-making payoff. Both papers focus on approximate truthfulness of a calibration measure in the online setting, while our paper focuses on the perfect truthfulness in the batch setting. Hartline et al. (2025) design a perfectly truthful calibration measure for binary classification in the batch setting. Our work generalizes Hartline et al. (2025) beyond binary outcomes to multi-class classification tasks. We also provide empirical evaluations of the truthful error metric.

**Calibration for Multi-Class Prediction.** Calibration in multi-class settings has been studied from several perspectives. Guo et al. (2017) introduced confidence calibration methods such as temperature scaling. Minderer et al. (2021) empirically observe that the ranking by calibration error is not robust to hyperparameter (binning size) selection, which motivated our paper. Kull et al. (2019) discusses the distinction between confidence calibration and class-wise calibration. Gopalan et al. (2024) studies projected smooth calibration, reducing a multi-class calibration measure to a binary-class smooth calibration measure. Zhao et al. (2021); Tang et al. (2025) studies decision calibration, which calibrates predictions relative to downstream decision tasks. These papers do not study truthfulness, which is the focus of our work.

## 2 PRELIMINARIES

Let $\mathbb{I}[E]$ be the indicator function, which equals 1 if statement $E$ is true and 0 otherwise. We use $\Delta(\mathcal{Y})$ to denote the set of probability distributions on the outcome space $\mathcal{Y} = \{1, \ldots, k\}$. We view each distribution $p \in \Delta(\mathcal{Y})$ equivalently as a vector $p = (p_1, \ldots, p_k) \in \mathbb{R}^k$ where $p_r$ is the probability mass $\Pr_{y \sim p}[y = r]$ for every $r = 1, \ldots, k$.

We consider a $k$-class classification task. Each sample $(x, y) \sim D$ is drawn independently and identically from a distribution $D$. A sample consists of a feature $x$ from the feature space $X$ and an outcome $y$ from the outcome space $\mathcal{Y} = \{1 \ldots k\}$. Given feature $x \in X$, the goal of a predictor $f$ is to output a probabilistic prediction $f(x) \in \Delta(\mathcal{Y})$. We use $f_r(x)$ to denote the $r$-th coordinate of $f(x)$, i.e., the predicted probability of the outcome $y$ being $r \in \mathcal{Y} = \{1, \ldots, k\}$.

A predictor is calibrated if its probabilistic predictions are conditionally correct.

**Definition 2.1** (Calibration for $k$-class prediction). *A predictor $f : X \to \Delta(\mathcal{Y})$ is (perfectly) calibrated if for every $p \in \Delta(\mathcal{Y})$ and every $r \in \mathcal{Y} = \{1, \ldots, k\}$,*

$$\Pr_{(x,y) \sim D}[y = r | f(x) = p] = p_r.$$

Definition 2.1 generalizes the notion of calibration from binary prediction to $k$-class prediction. In binary prediction, the outcome space is $\{0, 1\}$. A predictor $f : X \to [0, 1]$ maps a feature $x \in X$ to a (scalar) probability value $f(x) \in [0, 1]$. On a distribution $D$ of feature-outcome pairs $(x, y) \in X \times \{0, 1\}$, calibration for binary prediction is defined as follows:

**Definition 2.2** (Calibration for binary prediction). *A predictor $f : X \to [0,1]$ is (perfectly) calibrated if for every $p \in [0,1]$, $\Pr_{(x,y)\sim D}[y = 1|f(x) = p] = p$.*

Given a predictor $f$, we can quantify how miscalibrated it is using a calibration measure. Specifically, given $n$ data points $(x_1, y_1), \ldots, (x_n, y_n)$ drawn i.i.d. from the underlying distribution $D$, we apply our predictor $f$ to get the corresponding predictions $p_1 := f(x_1), \ldots, p_n := f(x_n)$. A calibration measure CAL maps the $n$ prediction-outcome pairs $(p_1, y_1), \ldots, (p_n, y_n)$ to a real-valued calibration error $\mathrm{CAL}(p_1, \ldots, p_n; y_1, \ldots, y_n) \in \mathbb{R}$.

We now introduce the notion of *truthfulness* for calibration measures (Haghtalab et al., 2024).

**Definition 2.3** (Truthfulness). *We say a calibration measure $\mathrm{CAL} : \Delta(\mathcal{Y})^n \times \mathcal{Y}^n \to \mathbb{R}$ is truthful if for all choices of ground-truth distributions $p_1^*, \ldots, p_n^* \in \Delta(\mathcal{Y})$ and all alternative predictions $p_1, \ldots, p_n \in \Delta(\mathcal{Y})$,*

$$\mathbf{E}[\mathrm{CAL}(p_1^*, \ldots, p_n^*; y_1, \ldots, y_n)] \leq \mathbf{E}[\mathrm{CAL}(p_1, \ldots, p_n; y_1, \ldots, y_n)],$$

*where the expectations on both sides are over $y_i \sim p_i^*$ independently for every $i = 1, \ldots, n$.*

We define the completeness and soundness of a calibration measure in Appendix A.1.

## 2.1 Calibration Measures for Binary Prediction

A commonly used calibration measure for binary prediction is the Expected Calibration Error (ECE), defined as the average absolute bias of the predictions:

**Definition 2.4** (ECE). *Given $n$ prediction-outcome pairs $(p_1, y_1), \ldots, (p_n, y_n) \in [0,1] \times \{0,1\}$, the Expected Calibration Error (ECE) is defined as*

$$\mathrm{ECE}(p_1, \ldots, p_n; y_1, \ldots, y_n) := \frac{1}{n} \sum_{i=1}^{n} |p_i - \bar{y}_{p_i}|,$$

*where $\bar{y}_p := \sum_{i=1}^{n} y_i \mathbb{I}[p_i = p] / \sum_{i=1}^{n} \mathbb{I}[p_i = p]$ is the average of $y_i$ conditioned on $p_i = p$.*

In the above definition, we use $\bar{y}_p$ to estimate the population conditional probability $\Pr_{(x,y)\sim D}[y = 1|f(x) = p]$. For this to be an accurate estimate, among the $n$ sampled prediction-outcome pairs $(p_i, y_i) = (f(x_i), y_i)$, there need to be enough pairs satisfying $p_i = p$ so that the sampling error can be reduced by taking the average over these pairs. In reality, however, it is common that all the $n$ sampled $p_i$'s are distinct, in which case $\bar{y}_p$ is equal to $y_i \in \{0,1\}$ for the *single $i$* satisfying $p_i = p$, which is not a good estimate for $\Pr_{(x,y)\sim D}[y = 1|f(x) = p] \in [0,1]$.

One popular way to reduce the sampling error is to use *binning* to group similar predictions together. We follow Minderer et al. (2021) to use quantile-based binning. Specifically, we first sort the prediction-outcome pairs $(p_1, y_1), \ldots, (p_n, y_n)$ so that the predictions are in increasing order: $p_1 \leq \cdots \leq p_n$. We then partition the indices $1, \ldots, n$ into $m$ consecutive bins $B_1, \ldots, B_m$, where

$$B_j = \left\{ i \in \{1, \ldots, n\} \,\middle|\, \frac{(j-1)n}{m} < i \leq \frac{jn}{m} \right\}, \quad \text{for } j = 1, \ldots, m. \tag{1}$$

This binning scheme ensures all bins have roughly equal size $|B_j| \approx n/m$. The number of bins, denoted by $m$, serves as a hyperparameter that can increase with the sample size $n$.

**Definition 2.5** (Quantile-binned ECE). *Let $(p_1, y_1), \ldots, (p_n, y_n) \in [0,1] \times \{0,1\}$ be $n$ prediction-outcome pairs. We sort these pairs in increasing order of the predictions: $p_1 \leq \cdots \leq p_n$. Let $m$ be a positive-integer hyperparameter specifying the number of bins we use, and define the bins $B_1, \ldots, B_m$ as in equation 1. In each bin $B_j$, we define the average prediction and average outcome as follows: $\bar{p}_j := \frac{1}{|B_j|} \sum_{i \in B_j} p_i, \bar{y}_j := \frac{1}{|B_j|} \sum_{i \in B_j} y_i$. The $\ell_1$ and $\ell_2$ quantile-binned ECE are defind as follows:*

$$\ell_1\text{-QECE}_m(p_1, \ldots, p_n; y_1, \ldots, y_n) := \sum_{j=1}^{m} \frac{|B_j|}{n} |\bar{p}_j - \bar{y}_j| = \frac{1}{n} \sum_{j=1}^{m} \left| \sum_{i=1}^{n} \mathbb{I}[i \in B_j] (p_i - y_i) \right|,$$

$$\ell_2\text{-QECE}_m(p_1, \ldots, p_n; y_1, \ldots, y_n) := \frac{1}{n^2} \sum_{j=1}^{m} \left( \sum_{i=1}^{n} \mathbb{I}[i \in B_j] (p_i - y_i) \right)^2.$$

It is a main result of Hartline et al. (2025) that $\ell_2\text{-QECE}_m$ is truthful:

**Theorem 2.6** (Hartline et al. (2025))**.** *In binary prediction, for every choice of the hyperparameter $m$, the calibration measure $\ell_2\text{-QECE}_m$ is truthful. Moreover, for every $p_1^*, \ldots, p_n^* \in [0, 1]$, assuming $y_i \in \{0, 1\}$ is drawn from $\mathrm{Ber}(p_i^*)$ independently for every $i = 1, \ldots, n$, the expected error achieved by predicting the truth can be computed as follows:*

$$\mathbf{E}[\ell_2\text{-QECE}_m(p_1^*, \ldots, p_n^*; y_1, \ldots, y_n)] = \frac{1}{n^2} \sum_{i=1}^{n} p_i^*(1 - p_i^*).$$

### 2.2 Calibration Measures for Multi-Class Prediction

Given a prediction-outcome pair $(p, y) \in \Delta(\mathcal{Y}) \times \mathcal{Y}$ for a $k$-class prediction task with outcome space $\mathcal{Y} = \{1, \ldots, k\}$, we can decompose it into $k$ prediction-outcome pairs for binary prediction tasks. Specifically, for $r = 1, \ldots, k$, we define $y^{(r)} := \mathbb{I}[y = r] \in \{0, 1\}$ as the binary indicator of the true label $y$ being $r$, and define $p^{(r)} := p_r \in [0, 1]$ as the predicted probability of $y$ being $r$. Now for each $r = 1, \ldots, k$, we get a prediction-outcome pair $(p^{(r)}, y^{(r)}) \in [0, 1] \times \{0, 1\}$ for a binary prediction task.

This reduction from $k$-class prediction to binary prediction allows us to lift calibration measures from binary prediction to multi-class prediction. Specifically, we can measure calibration for $k$-class prediction by aggregating the calibration error across the $k$ induced binary prediction tasks. We describe two natural aggregation methods below.

Class-wise aggregation simply takes the average calibration error over the $k$ binary prediction tasks.

**Definition 2.7** (Class-wise aggregation)**.** *Let $(p_1, y_1) \ldots, (p_n, y_n) \in \Delta(\mathcal{Y}) \times \mathcal{Y}$ be $n$ prediction-outcome pairs. Given a calibration measure $\mathrm{CAL}$ for binary prediction, we define a $k$-class calibration measure $\mathrm{CAL}^{(\mathsf{classwise})}$ as follows:*

$$\mathrm{CAL}^{(\mathsf{classwise})}(p_1, \ldots, p_n; y_1, \ldots, y_n) := \frac{1}{k} \sum_{r=1}^{k} \mathrm{CAL}(p_1^{(r)}, \ldots, p_n^{(r)}; y_1^{(r)}, \ldots, y_n^{(r)}).$$

Confidence aggregation focuses only on the outcome $r$ with the largest predicted probability $p^{(r)}$. Specifically, given $p \in \Delta(\mathcal{Y})$, we define $r_p := \arg\max_{r \in \{1, \ldots, k\}} p^{(r)}$.

**Definition 2.8** (Confidence Aggregation)**.** *Let $(p_1, y_1) \ldots, (p_n, y_n) \in \Delta(\mathcal{Y}) \times \mathcal{Y}$ be $n$ prediction-outcome pairs. Given a calibration measure $\mathrm{CAL}$ for binary prediction, we define a $k$-class calibration measure $\mathrm{CAL}^{(\mathsf{conf})}$ as follows: $\mathrm{CAL}^{(\mathsf{conf})}(p_1, \ldots, p_n; y_1, \ldots, y_n) := \mathrm{CAL}(p_1^{(r_{p_1})}, \ldots, p_n^{(r_{p_n})}; y_1^{(r_{p_1})}, \ldots, y_n^{(r_{p_n})}).$*

### 2.3 Proper Losses and Dominance Between Predictors

While calibration measures $\mathrm{CAL}$ take $n$ prediction-outcome pairs $(p_1, y_1), \ldots, (p_n, y_n) \in \Delta(\mathcal{Y}) \times \mathcal{Y}$ as input, standard loss functions $\ell$ in machine learning are usually defined on a single prediction-outcome pair (and then averaged over all pairs in a dataset). By standard terminology, a loss function $\ell : \Delta(\mathcal{Y}) \times \mathcal{Y}$ satisfying the truthfulness condition in Definition 2.3 is called a *proper loss*: a loss function $\ell : \Delta(\mathcal{Y}) \times \mathcal{Y}$ is *proper* if for every $p^*, p \in \Delta(\mathcal{Y})$, $\mathbf{E}_{y \sim p^*}[\ell(p^*, y)] \leq \mathbf{E}_{y \sim p^*}[\ell(p, y)]$. Two widely-used proper losses are the log loss (a.k.a., cross-entropy loss), the Brier loss (a.k.a., squared error) and the classification error. We provide examples of proper losses in Appendix A.2.

Proper losses are a key concept in statistical decision theory (Gneiting & Raftery, 2007). Each proper loss $\ell$ corresponds to a decision problem, where $\ell(p, y)$ is the loss incurred by a decision maker who best responds to the prediction $p$ and receives outcome $y$. Conversely, for every decision problem, the loss incurred by a best-responding decision maker can be expressed using a proper loss. We provide a more detailed discussion of this relationship in Appendix B. Thus, if a predictor $f$ leads to lower expected decision loss than a predictor $g$ consistently for all decision problems, then $f$ must have lower expected loss than $g$ for all proper loses, and vice versa. This motivates our definition of dominance between predictors:

**Definition 2.9** (Dominance)**.** *Let $D$ be an underlying distribution of $(x, y) \in X \times \mathcal{Y}$. Given two predictors $f, g : \mathcal{X} \to \Delta(\mathcal{Y})$, we say $f$ dominates $g$ if $f$ achieves a lower or equal expected loss for every proper loss: $\mathbf{E}_{(x,y) \sim \mathcal{D}}[\ell(f(x), y)] \leq \mathbf{E}_{(x,y) \sim \mathcal{D}}[\ell(g(x), y)]$ for all proper losses $\ell$.*

Intuitively, a dominating predictor is closer to the ground-truth distribution and consistently more informative for decision making across all decision problems.

## 3   TRUTHFUL CALIBRATION MEASURES FOR MULTI-CLASS PREDICTION

We introduce a truthful calibration measure $\ell_2\text{-QECE}^{(\text{classwise})}$ for multi-class classification, obtained by lifting the truthful calibration measure $\ell_2\text{-QECE}$ (Theorem 2.6) from binary prediction to multi-class prediction via class-wise aggregation (Definition 2.7). In fact, we prove a general result (Theorem 3.1) showing that classwise aggregation always preserves the truthfulness of a calibration measure from binary prediction to multi-class prediction:

**Theorem 3.1.** *Let* CAL *be a truthful calibration measure for binary prediction. Then* $\text{CAL}^{(\text{classwise})}$ *is a truthful calibration measure for $k$-class prediction.*

Theorem 3.1 can be easily proved by the linearity of expectation, using the fact that class-wise aggregation is defined as the average calibration error for $r$ binary prediction tasks. We defer the formal proof to Appendix C.1.

Combining Theorem 3.1 with Theorem 2.6, we obtain a truthful calibration measure $\ell_2\text{-QECE}_m^{(\text{classwise})}$ for $k$-class prediction.

In contrast to class-wise aggregation, confidence aggregation (Definition 2.8) does not always preserve truthfulness. With confidence aggregation, the predictor can strategically manipulate the outcome receiving maximum predicted probability by reporting non-truthfully, which can result in a lower expected error than predicting the truth. In Section 4, we provide examples showing this observation:

**Observation 3.2.** *The calibration measure $\ell_2\text{-QECE}_m^{(\text{conf})}$ is* not *truthful for $k$-class prediction, despite $\ell_2\text{-QECE}_m$ being truthful for binary prediction.*

## 4   DOMINANCE PRESERVING PROPERTY OF TRUTHFUL MEASURES

We prove Theorem 4.1 below showing that our truthful calibration measure $\ell_2\text{-QECE}_m^{(\text{classwise})}$ has a desirable *dominance-preserving property*: if a calibrated predictor $f_1$ dominates another calibrated predictor $f_2$ (w.r.t. all proper losses, as in Definition 2.9), then $f_1$ must have smaller or equal expected error when measured using $\ell_2\text{-QECE}_m^{(\text{classwise})}$ on a dataset of arbitrary size. This dominance-preserving property makes proper losses widely applicable (Gneiting, 2011).[3]

This dominance-preserving property holds regardless of the hyperparameter $m$ (number of bins) used to compute $\ell_2\text{-QECE}_m^{(\text{classwise})}$. Thus, it provides a theoretical justification for the robustness of $\ell_2\text{-QECE}_m^{(\text{classwise})}$, which we empirically observe in Figure 1 and Section 5: there is a clear positive correlation between the calibration errors and proper losses among different predictors, which persists across different choices of $m$.

We provide concrete examples in Table 1 contrasting the truthful $\ell_2\text{-QECE}_m^{(\text{classwise})}$ with other non-truthful measures, showing that the non-truthful calibration measures do not have the desired properties of dominance-reserving and robustness. While Theorem 4.1 proves the dominance-preserving property of the truthful $\ell_2\text{-QECE}_m^{(\text{classwise})}$ only among calibrated predictors, our examples show that the property approximately extends to mildly miscalibrated predictors as well.

**Theorem 4.1** (Dominance Preserving). *Let $D$ be a distribution of $(x, y) \in X \times \mathcal{Y}$, and let $m, n$ be arbitrary positive integers. Given predictor $f : X \to \Delta(\mathcal{Y})$, for $(x_1, y_1), \ldots, (x_n, y_n)$ drawn i.i.d. from $D$, define $\ell_2\text{-QECE}_m^{(\text{classwise})}(f) := \mathbf{E}\left[\ell_2\text{-QECE}_m^{(\text{classwise})}(f(x_1), \ldots, f(x_n); y_1, \ldots, y_n)\right]$. Then for every pair of calibrated predictors $f_1, f_2$ such that $f_1$ dominates $f_2$ (Definition 2.9),*

$$\ell_2\text{-QECE}_m^{(\text{classwise})}(f_1) \leq \ell_2\text{-QECE}_m^{(\text{classwise})}(f_2).$$

Theorem 4.1 follows immediately from Lemma C.1 in Appendix C.2, which shows that $\ell_2\text{-QECE}_m^{(\text{classwise})}(f)$ of a calibrated predictor $f$ is equal to its expected Brier loss multiplied by a fixed constant independent of $f$. Since the multiclass Brier loss is proper, dominance of $f_1$ over $f_2$ implies that $f_1$ has a smaller expected Brier loss, and thus a smaller $\ell_2\text{-QECE}_m^{(\text{classwise})}(f)$.

---

[3]Dominance-preserving is a property of proper losses by definition, but it is not straightforward that they apply to a truthful calibration measure. A calibration measure jointly evaluates $n > 1$ samples instead of 1 sample by proper losses.

| | $\ell_1$-QECE$_m^{(\text{conf})}$ | $\ell_1$-QECE$_m^{(\text{classwise})}$ | $\ell_2$-QECE$_m^{(\text{conf})}$ | $\ell_2$-QECE$_m^{(\text{classwise})}$ |
|---|---|---|---|---|
| $f_1$ | $\Theta\left(\sqrt{(1-\varepsilon_1)\varepsilon_1 \cdot \frac{m}{n}}\right)$ | $\Theta\left(\frac{1}{k}\sqrt{(1-\varepsilon_1)\varepsilon_1 \cdot \frac{m}{n}}\right)$ | $\frac{1}{n}\cdot(1-\varepsilon_1)\varepsilon_1$ | $\frac{2}{kn}(1-\varepsilon_1)\varepsilon_1$ |
| $f_2$ | $\varepsilon_2 +$ above | $\varepsilon_2/k +$ above | $\varepsilon_2^2/m +$ above | $2\varepsilon_2^2/km+$above |
| $f_3$ | $\Theta\left(\sqrt{(1-\frac{1}{k})\frac{1}{k} \cdot \frac{m}{n}}\right)$ | $\Theta\left(\frac{1}{k}\sqrt{(1-\frac{1}{k})\frac{1}{k} \cdot \frac{m}{n}}\right)$ | $\frac{1}{n}\cdot(1-\frac{1}{k})\frac{1}{k}$ | $\frac{1}{kn}(1-\frac{1}{k})$ |
| $f_4$ | $(k-1)\varepsilon_3+$above | $\frac{2(k-1)\varepsilon_3}{k}+$above | $\frac{(k-1)^2\varepsilon_3^2}{m}+$above | $\frac{(k-1)\varepsilon_3^2}{m}+$above |

Table 1: The expected error of the four predictors $f_1, f_2, f_3, f_4$ under different calibration measures.

**Numerical Examples.** We provide examples of four predictors that concretely explain the ranking by truthful and non-truthful calibration measures. All four predictors are defined on the same underlying distribution $D$ of $(x, y) \in X \times \mathcal{Y} = \{1, \ldots, k\}$ chosen as follows. The marginal distribution of $x$ is uniform over $X = \{1, \ldots, k\}$. Conditioned on $x$, the outcome $y$ is distributed as follows: with probability $1 - \varepsilon_1$ we have $y = x$, and with the remaining probability $\varepsilon_1$ we have $y = x + 1 \pmod{k}$. Here and in the following, the error parameters $\varepsilon_1, \varepsilon_2, \varepsilon_3$ are small positive real numbers.

The first predictor we consider is the groud-truth predictor $f_1 : X \to \Delta(\mathcal{Y})$. That is,

$$f_1(x) = (\underbrace{0, \ldots, 0}_{x-1}, 1 - \varepsilon_1, \varepsilon_1, \underbrace{0, \ldots, 0}_{k-x-1}) \text{ for } x = 1, \ldots, k-1; \quad f_1(k) = (\varepsilon_1, 0, \ldots, 0, 1 - \varepsilon_1).$$

The second predictor $f_2$ is a slightly biased version of $f_1$, obtained by replacing $1-\varepsilon_1$ with $1-\varepsilon_1-\varepsilon_2$ and replacing $\varepsilon_1$ with $\varepsilon_1 + \varepsilon_2$ in the definition of $f_1$ above.

The third predictor $f_3$ is the constant predictor: $f_3(x) = (1/k, \ldots, 1/k)$ for every $x \in X = \{1, \ldots, k\}$. While uninformative, the predictor $f_3$ is perfectly calibrated.

The fourth predictor $f_4$ is also a constant predictor and is a slightly biased version of $f_3$:

$$f_4(x) = (1/k + (k-1)\varepsilon_3, 1/k - \varepsilon_3, \ldots, 1/k - \varepsilon_3) \quad \text{for every } x \in X.$$

We compute the expected error $\mathbf{E}[\text{CAL}(f(x_1), \ldots, f(x_n); y_1, \ldots, y_n)]$ of all four predictors $f \in \{f_1, f_2, f_3, f_4\}$ on $n$ i.i.d. examples $(x_1, y_1), \ldots, (x_n, y_n)$ drawn from $D$. The result is shown in Table 1 for the four calibration measures below, where only the last one $\ell_2$-QECE$_m^{(\text{classwise})}$ is truthful:

$$\text{CAL} \in \{\ell_1\text{-QECE}_m^{(\text{conf})}, \ell_1\text{-QECE}_m^{(\text{classwise})}, \ell_2\text{-QECE}_m^{(\text{conf})}, \ell_2\text{-QECE}_m^{(\text{classwise})}\}.$$

The following conclusions can be derived from Table 1:

1. The first three calibration measures are non-truthful, and they do not correctly reflect the dominance between the two calibrated predictors $f_1$ and $f_3$. This can be seen by the fact that the expected error of the **uninformative** $f_3$ is **lower** than the **ground-truth** predictor $f_1$ when $k \gg \frac{1}{\varepsilon_1}$ for the first three calibration measures. In particular, this confirms Observation 3.2.

2. Using the first three non-truthful calibration measures, the **miscalibrated** and **uninformative** constant predictor $f_4$ achieves **smaller expected error** than the **ground-truth** predictor $f_1$, when $k \gg 1/\varepsilon_1$ and $\varepsilon_3$ is sufficiently small.

3. **Truthful calibration measure is robust to hyperparameter selection.** As we prove in Section 3, the fouth calibration measure $\ell_2$-QECE$_m^{(\text{classwise})}$ is truthful: the ground-truth predictor $f_1$ achieves smaller or equal expected error than all other predictors (note that $2(1 - \varepsilon_1)\varepsilon_1 \leq 1/2 \leq 1 - 1/k$). When $\varepsilon_1 < 1/2$ and $\varepsilon_2$ is reasonably small, e.g., smaller than a sampling error $\varepsilon_2 = O(\frac{1}{\sqrt{n}})$, the slightly miscalibrated yet highly informative predictor $f_2$ achieves smaller expected error than the calibrated yet highly uninformative predictor $f_3$ under the truthful calibration measure $\ell_2$-QECE$_m^{(\text{classwise})}$, reflecting a desirable ranking between the two predictors. These results hold regardless of the total number $m$ of bins used when computing $\ell_2$-QECE$_m^{(\text{classwise})}$, demonstrating the robustness of this truthful calibration measure w.r.t. hyperparameter choice.

4. **Non-truthful calibration measure is less robust to hyperparameter selection.** Consider the three non-truthful calibration measures (first three columns of Table 1). Suppose $\varepsilon_1 \ll \frac{1}{k}$, in which case the uninformative $f_3$ has higher expected error than the ground-truth $f_1$. In the same regime as Item 3 above, when $\varepsilon_2 = O(1/\sqrt{n})$ is reasonably as small as the order of the sampling

error, and when $m$ is small (such as $m = O(1)$), the slightly miscalibrated but highly informative $f_2$ has a higher expected calibration error than the uninformative $f_3$. However, the order is reversed as $m$ increases. This observation explains the changing trends of ranking as $m$ changes in our empirical evaluations in Section 5.2, demonstrating the non-robustness of non-truthful measures.

## 5 EMPIRICAL EVALUATIONS

In this section, we conduct empirical evaluations of neural network predictors with different calibration measures. We study the errors' robustness to binning size selection as an implication of the dominance-preserving property. To see this implication, when a truthful calibration measure preserves the dominance, the ranking between calibrated predictors is consistent across different binning size selections. When the predictors are sufficiently close to calibration, the truthful calibration measure remains robust to the selection of a binning size.

### 5.1 EXPERIMENTAL SETUP

**Dataset.** We use the `CIFAR-100` dataset, which we split into 45,000 training images, 5,000 validation images, and 10,000 test images. The validation set is mainly used to perform temperature scaling on model checkpoints.

**Model.** We select common models with increasing parameter sizes, including MobileNetV3-Small, ResNet10t, ResNet18, ResNet34, ResNet50, and BiT-ResNetV2-50x1, which are all pretrained on `ImageNet` and then fine-tuned on the `CIFAR-100` dataset. During fine-tuning, we keep each checkpoint from the training process. For evaluation, we apply temperature scaling for all checkpoints, following Minderer et al. (2021). Specifically, the temperature of the model output is selected to minimize the cross-entropy on the validation set.

This section displays the plots with MobileNetV3 and the plots with all models. All plots for other models are deferred to Appendix E.

**Evaluation Metrics.** We report the log loss, the Brier loss (a.k.a squared loss), the classification error, the spherical loss, as well as the four calibration errors considered in Table 1 ($\ell_1$-QECE$^{(\text{conf})}$, $\ell_2$-QECE$^{(\text{conf})}$, $\ell_1$-QECE$^{(\text{classwise})}$, $\ell_2$-QECE$^{(\text{classwise})}$). We also tested fixed-binned ECE ($\ell_1$-ECE$^{(\text{conf})}$, $\ell_2$-ECE$^{(\text{conf})}$, $\ell_1$-ECE$^{(\text{classwise})}$, $\ell_2$-ECE$^{(\text{classwise})}$). These fixed-binned ECE are all non-truthful and we deferred these plots to Appendix E.

### 5.2 PER-MODEL CALIBRATION-LOSS TRADEOFFS

For each model in our suite, we plot the calibration-performance tradeoff by tracking checkpoints throughout training. Each dot in the plots corresponds to the performance of a checkpoint when training the model. The $x$-axis shows the log loss (a.k.a., cross-entropy loss) and the $y$-axis shows another proper loss or a calibration error metric. The color of each dot reflects the checkpoint epoch: earlier checkpoints use cooler colors (blue), while later checkpoints use warmer colors (red).

We show the results for MobileNetV3 in Figure 2 as an example. We note that, with a large number of bins, the ranking by a calibration error approaches the same by a loss function. For example, when $m = n$, the $\ell_2$-ECE is the same as the squared loss.

We have two main findings. First, in Figure 2a, the checkpoints reveal a consistent *dominance* relationship across tested proper losses: more accurate predictors (lower log loss) also achieve a lower classification error, squared loss, and spherical loss. This dominance across losses demonstrates that model quality can be meaningfully compared along the training trajectory.

Second, our truthful calibration error, the $\ell_2$-QECE (the third and fourth plot in Figure 2c) *preserves this dominance* between checkpoints. The classical binned ECE with confidence aggregation flips the ranking between predictors under different binning sizes. The truthful error preserves the dominance relationship: as the loss decreases, the calibration error also decreases. A truthful error ensures that conclusions drawn about which checkpoint or model is better calibrated remain robust to hyperparameter selections.

Moreover, as $m$ increases, an informative predictor with a lower log loss has a lower calibration error than the less informative predictors with a high log loss. This is predicted by the observation in Item 4 from Section 4.

## 5.3 CROSS-MODEL COMPARISONS

When aggregating all models and checkpoints into a single plot, we find that the same conclusion as in Section 5.2 holds even across different models: the dominance relationships between models are generally preserved with only a few exceptions. For all checkpoints of all models, we construct the largest superset that may admit a dominant total order.[4] In Figure 3, models in the dominance ordering are shown as colored dots, while the others are shown as gray dots. Among the predictors that have a dominance ordering, the truthful calibration measure preserves the ordering and is robust to binning size selection.

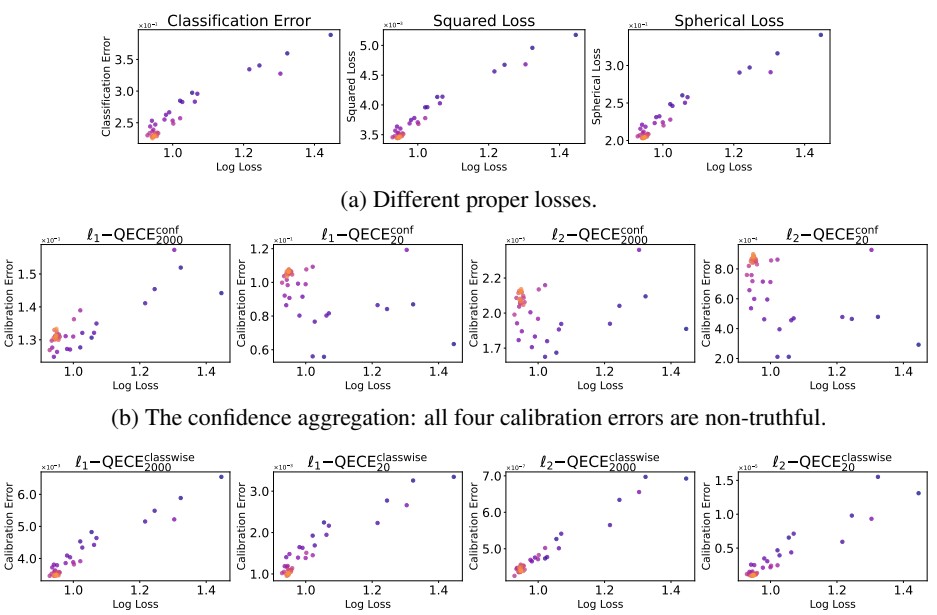

(a) Different proper losses.

(b) The confidence aggregation: all four calibration errors are non-truthful.

(c) The classwise aggregation: $\ell_2$-QECE$^{(\text{classwise})}$ in the third and fourth plots are truthful.

Figure 2: Calibration error and proper losses of different checkpoints of MobileNetV3 on the test set. Each dot in the plot corresponds to one checkpoint. The $x$-axis of each plot is the log loss. The $y$-axis shows a different calibration error / proper loss.

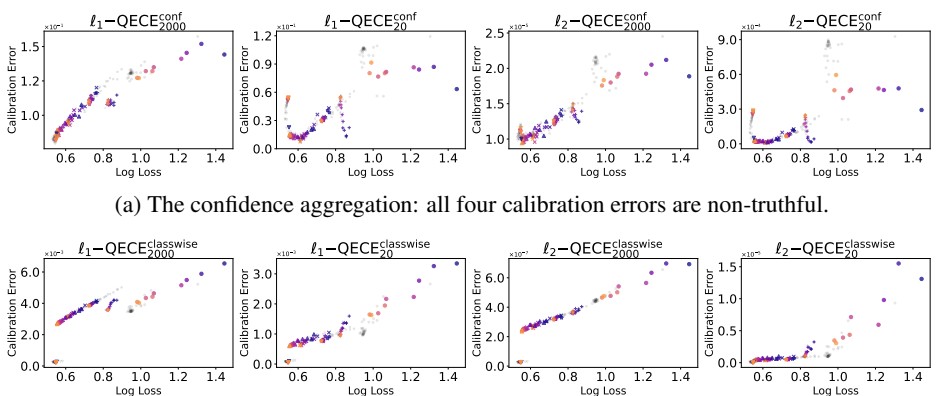

(a) The confidence aggregation: all four calibration errors are non-truthful.

(b) The classwise aggregation. $\ell_2$-QECE$^{(\text{classwise})}$ in the third and fourth plots are truthful.

Figure 3: Calibration errors and log loss of checkpoints for all models we evaluated. Each dot in the plot corresponds to a checkpoint of one neural network model. The $x$-axis of each plot is the log loss. The $y$-axis shows different calibration errors. We plot in colors the models in the maximal dominant total order and plot the rest of the models in grey.

---

[4]Specifically, if one model dominates another, all of its losses must be smaller. This defines a possible dominance partial order, from which we extract the longest path.

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

# A  ADDITIONAL PRELIMINARIES

## A.1  CALIBRATION MEASURE: COMPLETENESS AND SOUNDNESS

The two requirements for a calibration measure are *completeness* and *soundness*. Completeness and soundness ensure that the calibration error vanishes only for calibrated predictors and does not vanish for miscalibrated predictors. Note that proper losses are truthful, but are not calibration errors, i.e., they do not have completeness or soundness of a calibration measure.

**Definition A.1** (Completeness). *A calibration measure* CAL *is complete in the limit if, for any calibrated predictor $f$,*

$$\lim_{n \to \infty} \mathbf{E}\left[\text{CAL}((f(x_i))_{i=1}^n, (y_i)_{i=1}^n)\right] = 0.$$

**Definition A.2** (Soundness). *A calibration measure* CAL *is sound in the limit if, for any miscalibrated predictor $f$,*

$$\lim_{n \to \infty} \mathbf{E}\left[\text{CAL}((f(x_i))_{i=1}^n, (y_i)_{i=1}^n)\right] > 0.$$

In this paper, we focus on several weaker definitions of high-dimensional calibration. The classwise aggregation focus on classwise calibration (Kull et al., 2019). Instead of conditioning on the high-dimensional prediction, the classwise calibration requires unbiasedness conditioned on a prediction of a single dimension.

**Definition A.3** (Classwise Calibration (Kull et al., 2019)). *A predictor $f : X \to \Delta(\mathcal{Y})$ is (perfectly) classwise calibrated if for every $p \in [0, 1]$ and every $r \in \mathcal{Y} = \{1, \ldots, k\}$,*

$$\Pr_{(x,y) \sim D}[y = r | f_r(x) = p] = p.$$

The confidence aggregation is induced from the confidence calibration. The confidence calibration requires unbiasedness conditioned on the highest prediction.

**Definition A.4** (Confidence Calibration). *A predictor $f : X \to \Delta(\mathcal{Y})$ is (perfectly) confidence calibrated if for every $p \in [0, 1]$,*

$$\Pr_{(x,y) \sim D}[y = \arg\max_r f_r(x) | \max_r f_r(x) = p] = p.$$

## A.2  COMMONLY USED PROPER LOSSES

We define the Brier loss and the classification error here.

**Definition A.5** (Brier Loss). *The multi-class Brier loss is*

$$l_{\text{Brier}}(p, y) := \sum_{r=1}^k \left(p^{(r)} - \mathbb{I}[y = r]\right)^2.$$

**Definition A.6** (Classification Error). *The multi-class classification error is*

$$l_{\text{Cla}}(p, y) := \mathbb{I}\left[y \neq \arg\max_{r \in \{1, \ldots, k\}} p_r\right].$$

# B  DECISION-THEORETIC JUSTIFICATION OF DOMINANCE

In this section, we provide the decision-theoretic justification of dominance between predictors by proper losses in Section 2.3. We will show that proper losses admit an interpretation through the classical statistical decision theory. Every proper loss corresponds to the Bayes risk of following a prediction for some decision task.

**Definition B.1** (Decision problem). *A (finite) decision problem consists of an outcome space $\mathcal{Y}$, an action space $A$, and a loss*

$$L : A \times \mathcal{Y} \to \mathbb{R}_{\geq 0}, \qquad (a, y) \mapsto L(a, y).$$

*Given a predictive distribution $q \in \Delta(\mathcal{Y})$, the* Bayes risk *of an action $a \in A$ under $q$ is*

$$R_L(a; q) := \mathbf{E}_{y \sim q}\big[L(a, y)\big].$$

*A* best response *to $q$ is any action*

$$BR_L(q) \in \arg\min_{a \in A} R_L(a; q),$$

*and the best-responding loss, which we will refer to as the induced loss is a function of the prediction and the outcome*

$$l_L(q, y) := L\big(BR_L(q), y\big), \qquad (q, y) \in \Delta(\mathcal{Y}) \times \mathcal{Y}.$$

A predictor that outputs $q$ induces a downstream decision $a = \mathrm{BR}_L(q)$; the realized loss when outcome $y$ occurs is precisely $l_L(q, y)$.

We restate the definition of a proper loss here.

**Definition B.2** (Proper loss)**.** *A loss $l : \Delta(\mathcal{Y}) \times \mathcal{Y} \to \mathbb{R}_{\geq 0}$ is proper if, for every $p \in \Delta(\mathcal{Y})$,*

$$\mathbf{E}_{y \sim p}\big[l(p, y)\big] \leq \mathbf{E}_{y \sim p}\big[l(q, y)\big] \quad \textit{for all } q \in \Delta(\mathcal{Y}).$$

The theorem below says that the space of proper losses is the same as the space of induced losses from decision problems.

**Theorem B.3** (Induced Losses $\Leftrightarrow$ Proper Losses)**.** *Induced losses and proper losses are equivalent in the following sense:*

- *For any decision problem $(A, \mathcal{Y}, L)$, the induced loss $l_L(q, y) = L(BR_L(q), y)$ is proper.*

- *Conversely, for any proper loss $l$, there exists a decision problem $(A, \mathcal{Y}, L)$ whose induced loss equals $l$; in particular, one can take $A = \Delta(\mathcal{Y})$ and $L(a, y) = l(a, y)$, so that $l_L = l$. If $l$ is strictly proper, then $BR_L(p) = \{p\}$ for all $p$.*

*Proof.* We only need to prove the induced loss is proper. Fix $p \in \Delta(\mathcal{Y})$ and any $q \in \Delta(\mathcal{Y})$. By definition of Bayes risk and best response,

$$\mathbf{E}_{y \sim p}\big[l_L(q, y)\big] = \mathbf{E}_{y \sim p}\Big[L\big(\mathrm{BR}_L(q), y\big)\Big] = R_L\big(\mathrm{BR}_L(q); p\big)$$

$$\geq \min_{a \in A} R_L(a; p) = R_L\big(\mathrm{BR}_L(p); p\big) = \mathbf{E}_{y \sim p}\big[l_L(p, y)\big],$$

so $l_L$ is proper.

$\square$

**Implication for dominance.** Given predictors $f, g : \mathcal{X} \to \Delta(\mathcal{Y})$ and any data distribution over $(X, Y)$, evaluating under *all* proper losses is equivalent to evaluating under *all* decision problems via Theorem B.3. Hence, the dominance

$$\mathbf{E}[l(f(X), Y)] \leq \mathbf{E}[l(g(X), Y)] \quad \text{for all proper } l,$$

is exactly *decision-theoretic dominance*: $f$ attains no larger Bayes risk than $g$ across all downstream decision problems. For calibrated predictors, this dominance is the Blackwell ordering in the economics literature (Blackwell, 1951).

## C  DEFERREF PROOFS

### C.1  PROOF OF THEOREM 3.1

*Proof.* Let $p_1^*, \ldots, p_n^* \in \Delta(\mathcal{Y})$ be ground-truth distributions, and let $p_1, \ldots, p_n \in \Delta(\mathcal{Y})$ be arbitrary predictions. Draw $y_i \in \mathcal{Y}$ from $p_i^*$ independently for $i = 1, \ldots, n$. This implies $y_i^{(r)} \sim \mathsf{Ber}((p_i^*)^{(r)})$ for every $r = 1, \ldots, k$. Since CAL is truthful, we have

$$\mathbf{E}[\mathrm{CAL}((p_1^*)^{(r)}, \ldots, (p_n^*)^{(r)}; y_1^{(r)}, \ldots, y_n^{(r)})] \leq \mathbf{E}[\mathrm{CAL}(p_1^{(r)}, \ldots, p_n^{(r)}; y_1^{(r)}, \ldots, y_n^{(r)})].$$

Summing up the above inequality over $r = 1, \ldots, k$ and applying the linearity of expectation, we get the truthfulness of $\mathrm{CAL}^{(\mathsf{classwise})}$:

$$\mathbf{E}[\mathrm{CAL}^{(\mathsf{classwise})}(p_1^*, \ldots, p_n^*, y_1, \ldots, y_n)] \leq \mathbf{E}[\mathrm{CAL}^{(\mathsf{classwise})}(p_1, \ldots, p_n, y_1, \ldots, y_n)]. \qquad \square$$

## C.2 PROOF OF THEOREM 4.1

Theorem 4.1 is a direct consequence of the following lemma:

**Lemma C.1.** *Let $D$ be a distribution of $(x, y) \in X \times \mathcal{Y}$, and let $f : X \to \Delta(\mathcal{Y})$ be a calibrated predictor. Let $m, n$ be arbitrary positive integers. For $(x_1, y_1), \dots, (x_n, y_n)$ drawn i.i.d. from $D$,*

$$\mathbf{E}\left[\ell_2\text{-QECE}_m^{(\text{classwise})}(f(x_1), \dots, f(x_n); y_1, \dots, y_n)\right] = \frac{1}{kn}\,\mathbf{E}_{(x,y)\sim D}\left[l_{\text{Brier}}(f(x), y)\right].$$

We first prove a binary version of Lemma C.1:

**Lemma C.2.** *Let $D$ be a distribution of $(x, y) \in X \times \{0, 1\}$, and let $f : X \to [0, 1]$ be a calibrated predictor. Let $m, n$ be arbitrary positive integers. For $(x_1, y_1), \dots, (x_n, y_n)$ drawn i.i.d. from $D$,*

$$\mathbf{E}\left[\ell_2\text{-QECE}_m(f(x_1), \dots, f(x_n); y_1, \dots, y_n)\right] = \frac{1}{n}\,\mathbf{E}_{(x,y)\sim D}\left[(f(x) - y)^2\right].$$

*Proof.* Since $f$ is calibrated, when conditioned on $f(x)$, the distribution of $y$ is $\text{Ber}(f(x))$, whose variance is $f(x)(1 - f(x))$. Thus

$$\mathbf{E}_{(x,y)\sim D}[(f(x) - y)^2] = \mathbf{E}[f(x)(1 - f(x))].$$

Similarly, the $n$ pairs $(f(x_1), y_1), \dots, (f(x_n), y_n)$ are distributed identically and independently, and when conditioned on each $f(x_i)$, the distribution of $y_i$ is $\text{Ber}(f(x_i))$. Thus by Theorem 2.6,

$$\mathbf{E}\left[\ell_2\text{-QECE}_m((f(x_i))_{i=1}^n; (y_i)_{i=1}^n)\right] = \frac{1}{n^2}\mathbf{E}\left[\sum_{i=1}^n f(x_i)(1 - f(x_i))\right] = \frac{1}{n}\mathbf{E}[f(x)(1 - f(x))],$$

where the last equality holds because $(x_1, y_1), \dots, (x_n, y_n)$ are drawn i.i.d. from $D$. Combining the two equations above proves the lemma. $\square$

*Proof of Lemma C.1.* By Lemma C.2, for every $r = 1, \dots, k$,

$$\mathbf{E}[\ell_2\text{-QECE}_m(f(x_1)^{(r)}, \dots, f(x_n)^{(r)}; y_1^{(r)}, \dots, y_n^{(r)})] = \frac{1}{n}\mathbf{E}_{(x,y)\sim D}[(f(x)^{(r)} - y^{(r)})^2].$$

The proof is completed by summing up the above equation over $r = 1, \dots, k$ and applying the linearity of expectation. $\square$

# D   EXPECTED ERRORS FOR THE FOUR PREDICTORS IN TABLE 1

We write the calculation of the four quantities $\ell_1\text{-QECE}^{(\text{conf})}$, $\ell_1\text{-QECE}^{(\text{classwise})}$, $\ell_2\text{-QECE}^{(\text{conf})}$, and $\ell_2\text{-QECE}^{(\text{classwise})}$ in this section. Throughout, we use the facts that (i) for *calibrated* binary predictors, the expected binned absolute calibration error scales as $\Theta\!\left(\sqrt{p(1 - p)}\,\sqrt{m/n}\right)$ and the truthful squared version equals $\frac{1}{n}\,p(1 - p)$ (Theorem 2.6); and (ii) an *additive* prediction bias of $\delta$ contributes $\delta$ to $\ell_1\text{-QECE}$ and $\delta^2/m$ to $\ell_2\text{-QECE}$ (per binary reduction), with classwise averaging introducing the appropriate $1/k$ factor and the number of active coordinates.

## D.1   PREDICTOR 1 (INFORMATIVE, CALIBRATED)

Each sample has two active coordinates with probabilities $(1 - \varepsilon_1,\ \varepsilon_1)$, randomly permuted; the binary reductions are calibrated with $p \in \{1 - \varepsilon_1, \varepsilon_1\}$ on the active coordinate and $p = 0$ on the inactives.

**Confidence.**   The confidence projection selects the larger of the two active coordinates; the binary task has $p = 1 - \varepsilon_1$, zero bias, and variance $p(1 - p) = (1 - \varepsilon_1)\varepsilon_1$. Hence

$$\ell_1\text{-QECE}^{(\text{conf})} = \Theta\!\left(\sqrt{(1 - \varepsilon_1)\varepsilon_1}\,\sqrt{\tfrac{m}{n}}\right),$$

$$\ell_2\text{-QECE}^{(\text{conf})} = \tfrac{(1 - \varepsilon_1)\varepsilon_1}{n}.$$

**Classwise.** Only two coordinates are ever nonzero; classwise averages over $k$ coordinates, so we pick up a factor $2/k$:

$$\ell_1\text{-QECE}^{(\text{classwise})} = \tfrac{2}{k}\,\Theta\!\Big(\sqrt{(1-\varepsilon_1)\varepsilon_1}\,\sqrt{\tfrac{m}{n}}\Big),$$

$$\ell_2\text{-QECE}^{(\text{classwise})} = \tfrac{2}{k}\cdot\tfrac{(1-\varepsilon_1)\varepsilon_1}{n}.$$

### D.2 PREDICTOR 2 (INFORMATIVE, MISCALIBRATED BY $\varepsilon_2$)

Same geometry as Predictor 1, but each active binary reduction is shifted by an additive bias of size $\varepsilon_2$ (on confidence) and of size $\varepsilon_2$ for each of the two active classwise tasks.

**Confidence.** Constant binwise absolute bias $\varepsilon_2$ (plus sampling), and squared bias $\varepsilon_2^2$ spread over $m$ bins:

$$\ell_1\text{-QECE}^{(\text{conf})} = \varepsilon_2 + \Theta\!\Big(\sqrt{(1-\varepsilon_1)\varepsilon_1}\,\sqrt{\tfrac{m}{n}}\Big),$$

$$\ell_2\text{-QECE}^{(\text{conf})} = \tfrac{\varepsilon_2^2}{m} + \tfrac{(1-\varepsilon_1)\varepsilon_1}{n}.$$

**Classwise.** Two active coordinates averaged over $k$:

$$\ell_1\text{-QECE}^{(\text{classwise})} = \tfrac{2}{k}\,\varepsilon_2 + \tfrac{2}{k}\,\Theta\!\Big(\sqrt{(1-\varepsilon_1)\varepsilon_1}\,\sqrt{\tfrac{m}{n}}\Big),$$

$$\ell_2\text{-QECE}^{(\text{classwise})} = \tfrac{2}{k}\cdot\tfrac{\varepsilon_2^2}{m} + \tfrac{2}{k}\cdot\tfrac{(1-\varepsilon_1)\varepsilon_1}{n}.$$

### D.3 PREDICTOR 3 (INFORMATIVE, CALIBRATED UNIFORM)

Constant prediction $f(x) \equiv (1/k,\dots,1/k)$ with $Y$ uniform on $[k]$. Every binary reduction is calibrated with $p = 1/k$.

**Confidence.** Confidence picks $p = 1/k$ (ties broken arbitrarily but symmetrically); thus

$$\ell_1\text{-QECE}^{(\text{conf})} = \Theta\!\Big(\sqrt{\tfrac{1}{k}\big(1-\tfrac{1}{k}\big)}\,\sqrt{\tfrac{m}{n}}\Big),$$

$$\ell_2\text{-QECE}^{(\text{conf})} = \tfrac{1}{n}\cdot\tfrac{1}{k}\big(1-\tfrac{1}{k}\big).$$

**Classwise.** All $k$ coordinates contribute equally; classwise average leaves the variance term divided by $k$:

$$\ell_1\text{-QECE}^{(\text{classwise})} = \tfrac{2}{k}\,\Theta\!\Big(\sqrt{\tfrac{1}{k}\big(1-\tfrac{1}{k}\big)}\,\sqrt{\tfrac{m}{n}}\Big),$$

$$\ell_2\text{-QECE}^{(\text{classwise})} = \tfrac{1}{kn}\big(1-\tfrac{1}{k}\big).$$

### D.4 PREDICTOR 4 (UNINFORMATIVE, MISCALIBRATED BY $\varepsilon_3$)

Constant prediction with one coordinate high and the rest low: $\big(\tfrac{1}{k}+(k-1)\varepsilon_3,\ \tfrac{1}{k}-\varepsilon_3,\ \dots,\ \tfrac{1}{k}-\varepsilon_3\big)$, and $Y$ uniform.

**Confidence.** Confidence always selects the high coordinate; the true conditional frequency is $1/k$. Thus binwise absolute bias is $(k-1)\varepsilon_3$, and its square adds $(k-1)^2\varepsilon_3^2$ across $m$ bins:

$$\ell_1\text{-QECE}^{(\text{conf})} = (k-1)\varepsilon_3 + O\big(\sqrt{m/n}\big),$$

$$\ell_2\text{-QECE}^{(\text{conf})} = \tfrac{(k-1)^2\varepsilon_3^2}{m} + \tfrac{1}{n}\cdot\tfrac{1}{k}\big(1-\tfrac{1}{k}\big).$$

**Classwise.** For a fixed class $r$, the binary reduction has prediction $p_r = \frac{1}{k} + (k-1)\varepsilon_3$ for the projected coordinate and $p_r = \frac{1}{k} - \varepsilon_3$ otherwise, while $\Pr(Y = r) = 1/k$. The average absolute bias across the $k$ reductions equals $\frac{1}{k}\big((k-1)\varepsilon_3 + (k-1)\varepsilon_3\big) = \frac{2(k-1)}{k}\varepsilon_3$, and the average squared bias equals $\frac{1}{k}\big((k-1)\varepsilon_3^2 + (k-1)\varepsilon_3^2\big) = \frac{2(k-1)}{k}\varepsilon_3^2$; quantile binning turns squared bias into a $\varepsilon_3^2/m$ contribution and adds the same variance $1/(k)(1-1/k)/n$ term as in the calibrated uniform case. Hence

$$\ell_1\text{-QECE}^{(\text{classwise})} = \frac{2(k-1)}{k}\varepsilon_3 + O\big(\sqrt{m/n}\big),$$

$$\ell_2\text{-QECE}^{(\text{classwise})} = \frac{(k-1)\varepsilon_3^2}{m} + \frac{1}{kn}\Big(1 - \frac{1}{k}\Big).$$

# E ADDITIONAL EMPIRICAL RESULTS

In this section, we provide supplementary empirical results that complement the plots reported in the main text. Here we include the full set of plots across all evaluated models. Besides, we provide the plots using fix-binned ECE as the calibration error. The results are shown in Figure 4 (Cross model comparisons), Figure 5 (MobileNetV3-Small), Figure 6 (ResNet10t), Figure 7 (ResNet18), Figure 8, (ResNet34), Figure 9 (ResNet50), and Figure 10 (BiT-ResNetV2-50x1), and . The code for empirical results is uploaded with supplementary materials.

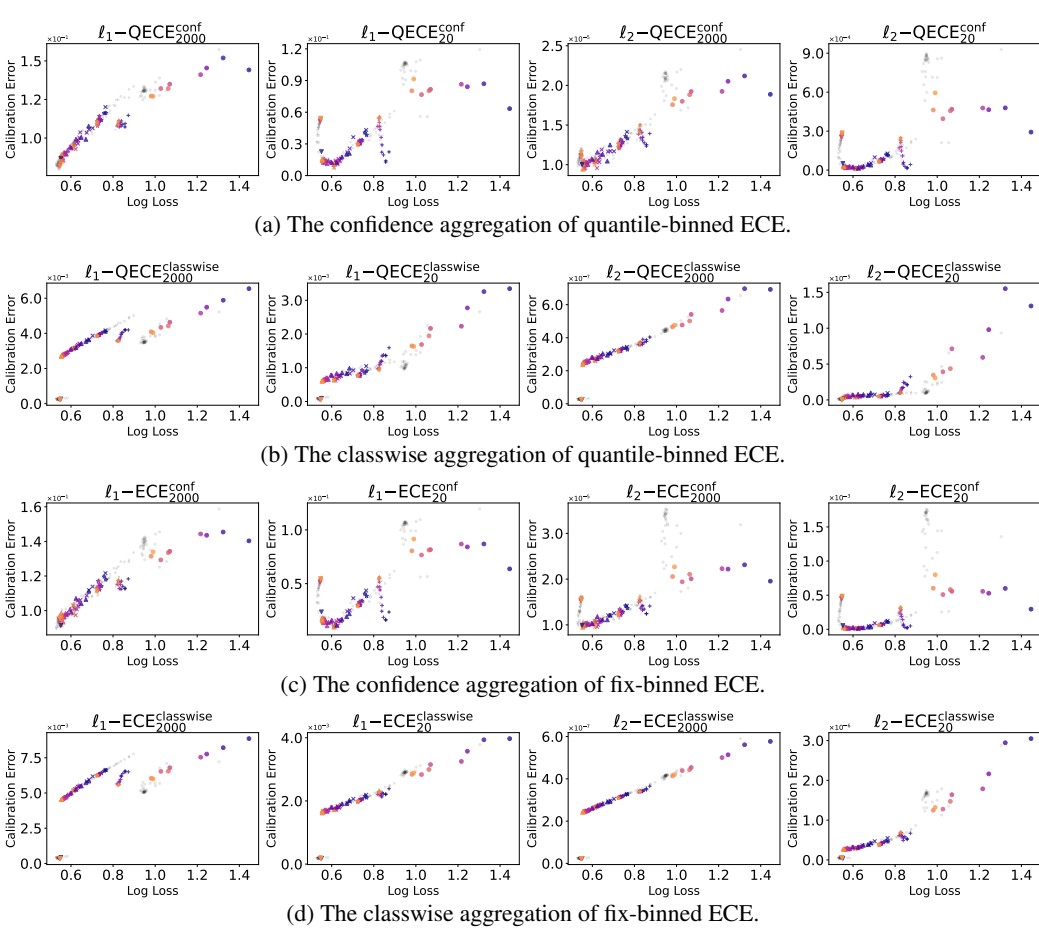

(a) The confidence aggregation of quantile-binned ECE.

(b) The classwise aggregation of quantile-binned ECE.

(c) The confidence aggregation of fix-binned ECE.

(d) The classwise aggregation of fix-binned ECE.

Figure 4: Calibration errors and log loss of checkpoints for all models we evaluated. Each dot in the plot corresponds to a checkpoint of one neural network model. The $x$-axis of each plot is the log loss. The $y$-axis shows different calibration errors. We plot in colors the models in the maximal dominant total order and plot the rest of the models in grey.

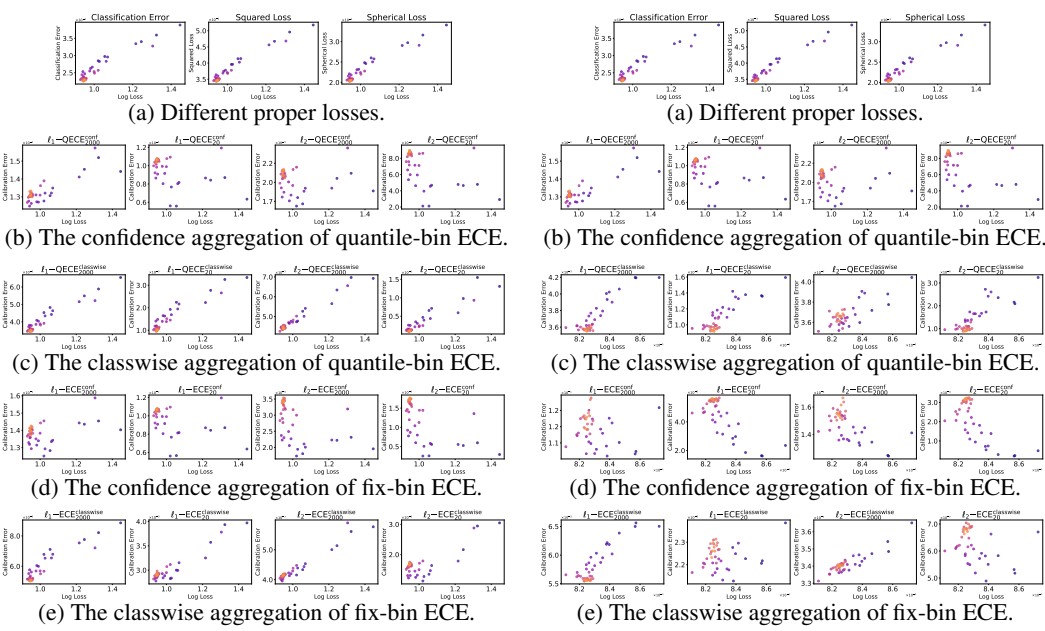

(a) Different proper losses.

(b) The confidence aggregation of quantile-bin ECE.

(c) The classwise aggregation of quantile-bin ECE.

(d) The confidence aggregation of fix-bin ECE.

(e) The classwise aggregation of fix-bin ECE.

Figure 5: Proper losses and calibration errors of (MobileNetV3-Small model.

(a) Different proper losses.

(b) The confidence aggregation of quantile-bin ECE.

(c) The classwise aggregation of quantile-bin ECE.

(d) The confidence aggregation of fix-bin ECE.

(e) The classwise aggregation of fix-bin ECE.

Figure 6: Proper losses and calibration errors of Resnet10t model.

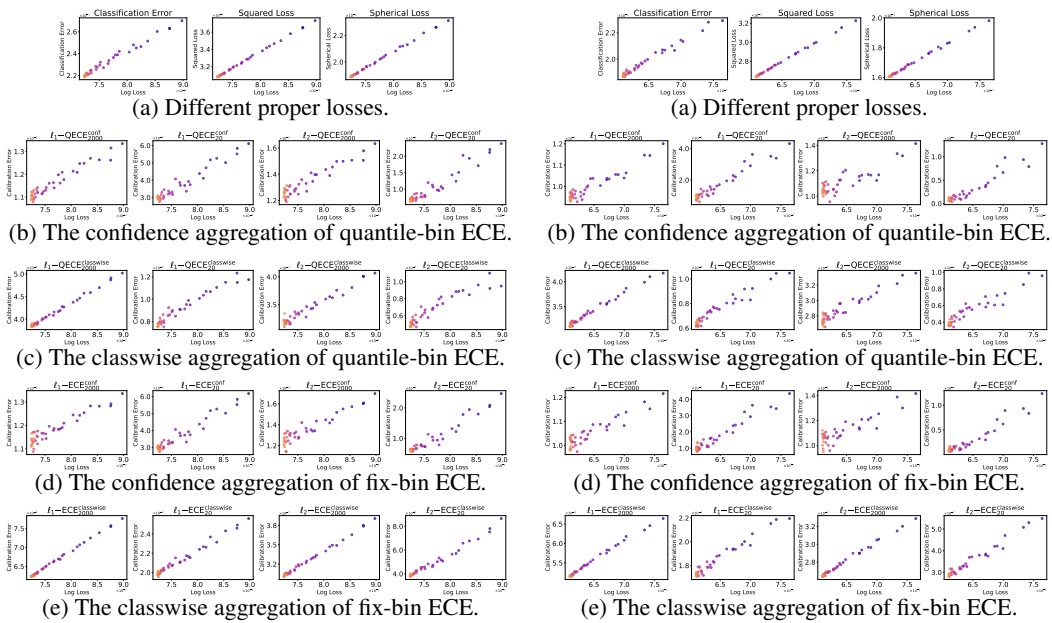

(a) Different proper losses.

(b) The confidence aggregation of quantile-bin ECE.

(c) The classwise aggregation of quantile-bin ECE.

(d) The confidence aggregation of fix-bin ECE.

(e) The classwise aggregation of fix-bin ECE.

Figure 7: Proper losses and calibration errors of Resnet18 model.

(a) Different proper losses.

(b) The confidence aggregation of quantile-bin ECE.

(c) The classwise aggregation of quantile-bin ECE.

(d) The confidence aggregation of fix-bin ECE.

(e) The classwise aggregation of fix-bin ECE.

Figure 8: Proper losses and calibration errors of Resnet34 model.

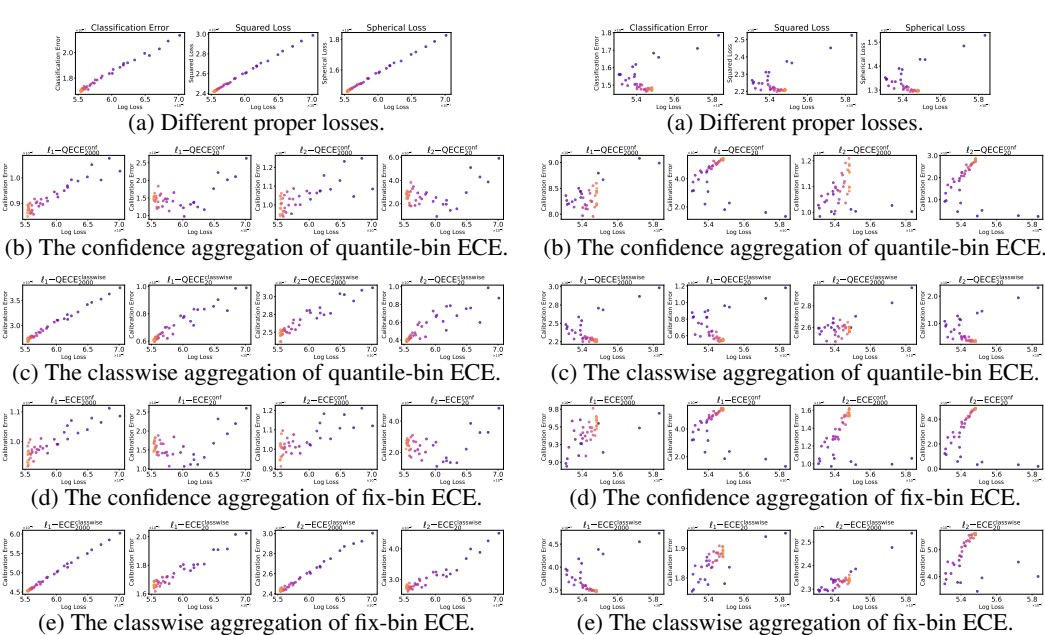

Figure 9: Proper losses and calibration errors of Resnet18 model.

Figure 10: Proper losses and calibration errors of BiT-ResNetV2-50x1 model.

