# OpenReview forum: "Making and Evaluating Calibrated Forecasts"
_ICLR.cc/2026/Conference — ICLR 2026 Conference Withdrawn Submission_

### Official Review · Reviewer_1hMP · 2025-10-31

**Soundness:** 3
**Presentation:** 3
**Contribution:** 3
**Rating:** 6
**Confidence:** 3

**Summary:**

This paper introduces a novel, theoretically grounded truthful calibration measure for multi-class prediction by extending existing binary truthful calibration measures. Unlike common approaches that aggregate binary subproblem errors using confidence-based methods, which are not generally truthful, the proposed class-wise aggregation preserves truthfulness when moving from binary to multi-class settings. The resulting measure is theoretically justified for robustness, as it maintains decision-task dominance between calibrated predictors regardless of hyperparameter choices. Finally, the authors provide an empirical evaluation across a wide range of models of different sizes for the image classification task.

**Strengths:**

1. The paper is well-structured, with a clear motivation and several examples throughout the text that enhance its overall clarity.
2. Each step of the paper is guided by a clear theoretical motivation, which strengthens the overall work.
3. The experimental evaluation clearly demonstrates the issues with non-truthful classifiers and calibration measures, while also highlighting the superiority of the proposed measure.

**Weaknesses:**

1. The paper lacks a conclusion section that summarizes the insights and contributions. Adding one would significantly improve readability.
2. The paper refers to Theorem 1.1 on line 51, but this theorem does not exist.
3. The notations $l_1$ and $l_2$ appear in Figure 1 without proper introduction or explanation.
4. The experimental evaluation could be strengthened by including additional tasks and domains.
5. Providing a practical algorithm for evaluating the proposed truthful calibration measures would further enhance the paper.

**Questions:**

1. Does the same observation hold in other domains, such as text classification tasks? Moreover, when dealing with LLMs, predictions are typically made over vocabularies of more than 60k classes. It would be interesting to explore whether the proposed approach can be applied to LLMs and what results can be expected.

---

> ### Author Response · Authors · 2025-11-20
>
> We thank the reviewer for their thoughtful feedback and suggestions. Yes, experiments on other domains (e.g., text) are a great direction for future research. We will include a discussion on that in future versions.

---

### Official Review · Reviewer_5AnH · 2025-11-02

**Soundness:** 3
**Presentation:** 3
**Contribution:** 2
**Rating:** 4
**Confidence:** 2

**Summary:**

This paper introduces a new truthful calibration measure for multi-class prediction tasks, extending recent work on truthful calibration from binary settings. Prior measures like Expected Calibration Error (ECE) are non-truthful, meaning they can incentivize predictors to misreport probabilities to appear more calibrated. Building on Hartline et al. (2025), who proposed a perfectly truthful binary calibration measure – l2-qECE(classwise), this paper generalizes it to the multi-class case using class-wise aggregation instead of confidence aggregation, proving that only the former preserves truthfulness. The authors further establish that this l2 classwise measure is robust, in contrast to other metrics considered.
Theoretically, the paper proves that truthful calibration errors preserve dominance among predictors. Empirically, experiments on CIFAR-100 with various neural networks confirm that this truthful measure preserves model rankings across different binning sizes, unlike non-truthful metrics whose rankings can flip.

**Strengths:**

- Extending truthful metrics to multi-class domain
- Theoretical contribution: calculated the estimates for some trivial predictors and proved the dominance preservation theorem, as well as showed that classwise aggregation preserve truthfulness
- The paper is well written

**Weaknesses:**

- Results are incremental. Novelty builds almost directly on Hartline et al. (2025) with a straightforward extension.

**Questions:**

-

---

> ### Author Response · Authors · 2025-11-20
>
> We thank the reviewer for their feedback. We clarify that our main contribution is the empirical findings, see our response above to all reviewers.

---

### Official Review · Reviewer_NXvi · 2025-11-03

**Soundness:** 4
**Presentation:** 2
**Contribution:** 2
**Rating:** 2
**Confidence:** 3

**Summary:**

This paper proposes a truthful calibration measure for multi-class prediction. This generalizes the measure proposed in Hartline et al for binary prediction. The method is empirically validated and shown to preserve relative performance of methods better than prior non-truthful approaches.

**Strengths:**

Calibration is an important subfield of machine learning, and having better metrics to measure calibration can improve our ability to study it empirically. The calibration metric proposed in this paper does seem to be better than prior metrics, at least in terms of the empirical evidence in the last section. It is also relatively straightforward to compute.

**Weaknesses:**

This paper overall seems like a very straightforward generalization of "A Perfectly Truthful Calibration Measure" by Hartline et al. (2025). I'm not sure that there's anything in here that is novel that would not occur to an expert on calibration reading that paper. I don't even think an expert would need to read that paper to prove Theorem 3.1; it just follows (as the paper says) from linearity of expectation.

It's also not clear to me how much interest this sort of truthful calibration is to the community. I don't think any paper on this subject has been published at ICLR (granted, it is a very new formalization).

**Questions:**

What do the authors see as the novel part of their contribution in this paper, relative to Hartline et al. (2025)?

---

> ### Author Response · Authors · 2025-11-20
>
> We thank the reviewer for their feedback. We clarify that our main contribution is the empirical findings, see our response above to all reviewers.

---

### Official Review · Reviewer_kYmS · 2025-11-03

**Soundness:** 4
**Presentation:** 4
**Contribution:** 2
**Rating:** 4
**Confidence:** 4

**Summary:**

This paper studies truthful calibration metrics as introduced by Haghtalab et al. (2024). A truthful calibration metric is one for which the ground truth $p^*$ always receives the best error out of all predictors, on IID samples from any distribution. The recent work of Hartline et al. (2025) gives a few truthful calibration measures for the setting of binary prediction.

This paper points out that via linearity of expectation, any truthful calibration measure for binary prediction can be extended to one for multiclass prediction. In particular, define a new calibration metric “classwise” as the average calibration error of each of the $k$ binary prediction tasks. Then, if a calibration measure is truthful in the binary case, it will be truthful for this new aggregated multiclass calibration measure. The paper points out that this is not the case for confidence calibration (i.e., the “top predicted probability class” calibration error).

The paper also defines a notion of a predictor “dominating” another predictor in terms of proper losses. In particular, $f$ dominates $g$ if $f$ has lower average loss than $g$ over all proper losses.

First, in section 3, the paper uses the fact that the multiclass “classwise” calibration error is truthful for any truthful binary calibration error in order to propose a new truthful multiclass calibration error: $\ell_2-QECE^{\text{classwise}}$. This is simply the $\ell_2$ calibration error over equal mass bins chosen via sorting the predictions by predicted probability.

In section 4, the paper demonstrates that if $f$ dominates $g$, then $f$ will have lower  $\ell_2-QECE^{\text{classwise}}$ calibration error than $g$. This follows simply from the fact that Brier score is a proper loss (and hence, $f$ has lower Brier score than $g$ by definition of domination), and that the proposed calibration error is essentially the Brier score. Section 4 also includes a simple counter-example which demonstrates that the proposed calibration measure has nice behavior even when other discussed measures fail.

Lastly, Section 5 has simple experiments on CIFAR-100 demonstrating that the truthfulness properties and robustness to bin size hold in practice. In particular, other calibration measures will flip the ranking of models with the same loss at different bin sizes, whereas the proposed truthful multiclass measure does not. Furthermore, the same holds even when comparing different models or model training checkpoints.

**Strengths:**

The paper is very clearly written. Although I am familiar with calibration, I had not yet got around to reading the truthful calibration line of work. I was easily able to understand the motivation and setup behind truthful calibration measures.

The main results are also very simple to understand. In fact, most follow as a corollary of the work Hartline et al. (2025). The experimental evidence / plots given are also convincing.

**Weaknesses:**

I am a bit unsure if the paper meets the content bar for ICLR. I would appreciate the opinion of other reviewers as well. In particular, from a theory perspective the results are extremely simple (almost all are 1-line proofs). In fact, the main contribution here could exist as essentially a page or two in Hartline et al. (2025), which, I will note, is a very recent paper (arxived August 18th 2025). Furthermore, the experiments, while covering many different models, are only evaluated on CIFAR-100.

I believe that the paper either needs 1) more theoretical results, perhaps relating dominance and truthful calibration (see below discussion for some potential ideas / discussions); or 2) more comprehensive empirical results, over multiple model types (classical, deep learning, language classification, LLM, etc.) or multiple datasets. I believe both would be welcome additions and improve the contribution content of the paper. As it stands, I don’t see this paper as more than a short note that the previous result from Hartline can extend in a simple manner. 2) may be especially useful to gain broader buy-in from the practical / empirical community, who may stand to benefit from a new simple calibration metric with clear benefits.

Nonetheless, I am open to discussion on this fact. I don’t believe that something being simple should be the reason to _not_ accept the paper. I just am unable to frame the scope of the contribution in the canon of truthful calibration literature.

I think the dominance vs. truthful calibration discussion can probably be fleshed out far more than it currently is. In particular, there are many works which point to correlation between calibration measures and accuracy in both classical ML (Chidambaram et al., 2024, Tao et al., 2024), multicalibration (Hansen et al., 2024) and more modern settings like LLMs (Mei et al., 2025). How does this discussion relate to the experiments conducted, and dominance vs. truthfulness more broadly? In particular, if we use calibration measures that correlate perfectly with accuracy / loss, are we even measuring anything interesting? Should we just be looking at the (proper) loss instead, and call it a day?

This all being said, I am still borderline because a simple idea does not imply that the paper should be rejected.

Mei et al., 2025. Reasoning about Uncertainty: Do Reasoning Models Know When They Don't Know?
Hansen et al., 2024. When is multicalibration post-processing necessary?
Chidambaram et al., 2024. Reassessing how to compare and improve the calibration of machine learning models.
Tao et al., 2024. A benchmark study on calibration.

Suggestions
1. I would at least mention the completeness and soundness requirements for calibration metrics before defining truthfulness. I was confused until I read the appendix A.1 which pointed out that proper losses do not satisfy these criteria. It would also be helpful, in A.1, to point out other calibration metric papers which have similar criteria / desiderata (for example, Błasiok et al. 2023, Hartline et al.).
2. I also have a bit of a bone to pick with the title. The title is extremely general, but the main result is really quite narrow. I believe a more suitable title would be something like “Multiclass Truthful Calibration Measures”, or something more along this line.

Błasiok et al. 2023. A Unifying Theory of Distance from Calibration.

Typos:
1. 691: “Deferref" -> Deferred
2. 308: “dominance-reserving” -> “dominance-preserving”

**Questions:**

To my understanding, the multiclass classwise aggregation scheme must recompute bins for each class in order to retain truthfulness guarantees. Is this the case?

---

> ### Author Response · Authors · 2025-11-20
>
> We thank the reviewer for their thoughtful feedback and suggestions. We clarify that our main contribution is the empirical findings, see our response above to all reviewers. For the specific question of the reviewer: 'To my understanding, the multiclass classwise aggregation scheme must recompute bins for each class in order to retain truthfulness guarantees. Is this the case?' Yes, that is correct.

---

### Author Response · Authors · 2025-11-18
**Clarification of our main contribution**

We thank the reviewers for their thoughtful feedback.

We believe there may have been a misunderstanding about the main contribution of our paper, perhaps due to the way we present the paper. The paper is not mainly about introducing a truthful calibration error for multi-class prediction.

Our **main result** is that truthful calibration errors are important for **empirical work** aiming to understand calibration. In particular, we show that non-truthfulness helps explain well-known issues where earlier calibration errors give different model rankings when hyperparameters (bin sizes) are changed. This empirical issue can thus be fixed by the truthful calibration error of Hartline et al (see Note * below). Before our work, it was not clear that this ranking inconsistency was caused by non-truthfulness. Prior work proposes non-constructive explanations, such as estimation bias or convergence rate to the limit [1, 2]. Our result demonstrates how important truthfulness is when evaluating the calibration of models. Since our main empirical result did not feature strongly in the reviews, we understand that we need to improve the paper's narrative so that this main message is clearer.

Note * : we had to generalize Hartline et al to multi-class setting, which is a **smaller contribution** in our paper but necessary. While the classwise extension may be a direct step from earlier work, the fact that standard practice (confidence aggregation) is not truthful is not obvious from prior work.


[1] Nixon, J., Dusenberry, M. W., Zhang, L., Jerfel, G., & Tran, D. (2019, June). Measuring calibration in deep learning. In CVPR workshops (Vol. 2, No. 7).

[2] Roelofs, R., Cain, N., Shlens, J., & Mozer, M. C. (2022, May). Mitigating bias in calibration error estimation. In International Conference on Artificial Intelligence and Statistics (pp. 4036-4054). PMLR.

---

> ### Comment · Reviewer_NXvi · 2025-11-25
>
> > the fact that standard practice (confidence aggregation) is not truthful is not obvious from prior work
>
> Isn't this the entire point of Haghtalab et al. (2024)? E.g. their abstract states "We conduct a taxonomy of existing calibration measures and their truthfulness. Perhaps surprisingly, we find that all of them are far from being truthful." If this is supposed to be the main contribution of the paper, why doesn't it fall under being just an empirical validation of results already shown theoretically by Haghtalab et al. (2024)?

---

> > ### Author Response · Authors · 2025-11-25
> >
> > We thank the reviewer for the opportunity for us to clarify.
> >
> > - Main empirical contribution: The reviewer mentioned our result as "an empirical validation of previous results". To our knowledge, none of the previous work connects robustness to truthfulness. There exists a previous theory literature on truthfulness, and a separate empirical literature that studies this robustness problem. It was unclear whether truthfulness mitigates the problem.
> >
> > - Minor theory contribution (difference to Haghtalab et al.). Previous theory literature on truthful calibration error observes that no **binary** calibration error is truthful, restricted to binary classification. Our observation is, **even if you have a truthful binary calibration error**, common practice that aggregates the binary error into multi-class error is **non-truthful**, e.g., the confidence aggregation.

---

### Comment · Area_Chair_dKGX · 2025-11-29

Dear Reviewers,

Authors’ kindly tried to address your concerns. If the responses address your concerns please acknowledge that. If not, please express remaining concerns. Thanks for your efforts!

Best, AC

---

### Note · Authors · 2025-12-01

I have read and agree with the venue's withdrawal policy on behalf of myself and my co-authors.